# A Systematic Review on the Local Wisdom of Indigenous People in Nature Conservation

**Azlan Abas \*** 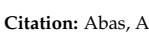**, Azmi Aziz and Azahan Awang**

Centre for Research in Development, Social and Environment (SEEDS), Faculty of Social Sciences and Humanities, Universiti Kebangsaan Malaysia, Bangi 43650, Selangor, Malaysia; abaz@ukm.edu.my (A.A.); azahan@ukm.edu.my (A.A.)
\* Correspondence: azlanabas@ukm.edu.my

**Abstract:** The local wisdom of indigenous people in nature conservation plays a critical part in protecting the planet's biodiversity and the overall health of the ecosystems. However, at the same time, indigenous people and their lands are facing immense threats through modernization and globalization. This study aims to systematically review and analyze the local wisdom of the indigenous people in nature conservation. The present study integrated multiple research designs, and the review was based on the published standard, namely the PRISMA statement (Preferred Reporting Items for Systematic Reviews and Meta-Analyses). This study used Web of Science (WoS) and Scopus as the main databases in searching for the required articles. Through content analysis, this study can be divided into seven main categories: (a) forest management, (b) flora and fauna conservation, (c) food security, (d) water management, (e) land management, (f) weather forecasting, and (g) others. The findings offer some basics on how academics can adopt and adapt the existing local wisdom of indigenous people in nature conservation into the scientific framework and design to answer the Sustainable Development 2030 Agenda.

**Keywords:** human ecology; indigenous people; sustainable development

## 1. Introduction

A kaleidoscopic diversity of the Earth's plants and animals underpins human existence but is under major threat from the environmental degradation wrought by human activities, from mining to agriculture [1]. A report by IPBES [2] has stated that nature is declining globally at rates unprecedented in mankind's history, and the rate of species extinctions is accelerating and will continue to do so if humans do not change the way they interact with nature. The human–nature conflict started with the beginning of human existence. Mankind is part of nature, but humans always see that they are at the top of the food chain or the end users of nature. In other words, it is the disposition of humans to consume natural resources at their will. However, due to this understanding and ideology, our planet is now ailing [3,4]. According to Abas et al. [5], recent trends in modernism are shifting from anthropocentrism towards technocentrism, but the change occurs at a very slow rate and with low impact. So, it has been suggested that modern society may need to look back and appreciate how the first people, better known as the indigenous people, took care of nature long before the arrival of modern civilization.

Over the decades, conservation by governments or conservation organizations has been regarded as the only legitimate form of conservation. In some cases, conservation of protected areas equaled the exclusion of the people, a view of conservation devoid of human components. However, not all human activities are anti-conservation [6,7]. For example, among the Kenyah people in the interior of North Kalimantan, in the Heart of the Borneo area, conservation and the use of natural resources amount to the same thing: to care for the forest as a source of livelihood, food, and good health, as well as cultural

identity [8,9]. There is a strong and deep bond between the community and the place. This in turn nourishes a belief that forest resources will continue to sustain the community in the future: if nature is respected, nature will give back and provide [10]. There is no clear distinction between conservation and livelihoods and culture; conservation is inclusive of those positive human components and the traditional knowledge and values of the communities that depend on those resources for a living and for their cultural survival. Indigenous conservation is inclusive and holistic [11].

Indigenous people have long histories with their land, which has provided sustenance in very direct and intimate ways. In addition, they tend to have a reciprocal relationship with nature, rather than viewing nature as existing to serve humans, as much of modern culture has historically regarded things [1,12,13]. The thinking, intuition, and practices of the indigenous people are called the local wisdom or local knowledge. Local wisdom is the basic knowledge gained from living in balance with nature. It is related to the culture in the community which is accumulated and passed on through the generations. This wisdom can be both abstract and concrete, but the important characteristics are that it comes from experiences or truth gained from life [14,15]. The local wisdom of indigenous people could be a critical part of protecting the planet's biodiversity and the overall health of ecosystems. Therefore, the participation of indigenous people in scientific assessments, policy planning, and roundtable discussions will be significant in finding a better way for sustainable development [16].

However, at the same time, the indigenous people and their lands are facing immense threats. They are dealing with pressures from encroaching infrastructure, agriculture, mining, logging, and other activities that also endanger biodiversity [17]. Besides that, the younger generation of the indigenous people, little by little, are leaving the practice of local wisdom due to modernization. This situation will eventually lead towards the loss of the local wisdom of the indigenous people which is valuable and precious for the protection of the biodiversity of nature [18].

There are several studies that have been conducted in recent years to explore the local wisdom of indigenous people from all over the world in nature conservation. However, the efforts to analyze all of the studies systematically are still lacking. Other than that, the trend and pattern of the study of the indigenous people's local wisdom in nature conservation also remain unanswered. Therefore, this study aims to systematically review and analyze the local wisdom of indigenous people in nature conservation. In order to achieve this aim, two research objectives were underlined: (1) to identify the local wisdom that has been practiced by the indigenous people in conserving nature and (2) to discover the patterns and differences of the local wisdom of the indigenous people in nature conservation among the papers studied.

## 2. Research Methodology

### 2.1. PRISMA Statement (Preferred Reporting Items for Systematic Reviews and Meta-Analyses)

This systematic review study used the PRISMA statement [19] as the main guideline for conducting the study. PRISMA is an evidence-based minimum list of monitoring elements for systematic analyses and meta-analysis. PRISMA emphasizes the reporting of randomized trial assessment reviews but can also be used as a basis for reporting comprehensive reviews of other forms of study, particularly intervention assessments [20].

### 2.2. Formulation of the Research Question

The formulation of the research questions was based on PICo. PICo is a tool that assists authors in developing suitable research questions for review. PICo is based on three main concepts, namely Population or Problem, Interest, and Context. Based on these concepts, this study has included three main aspects in the review: indigenous people (population), local wisdom (interest), and nature conservation (context). Therefore, this systematic review study asks the following questions: (1) what kind of local wisdom do indigenous people have for conserving nature? (2) What are the recommendations from

this study to encourage the adoption of the local wisdom of the indigenous people into the modern practice?

### 2.3. Systematic Searching Strategies

The systematic searching strategies involved three main strategies, which were identification, screening, and eligibility, as shown in Figure 1.

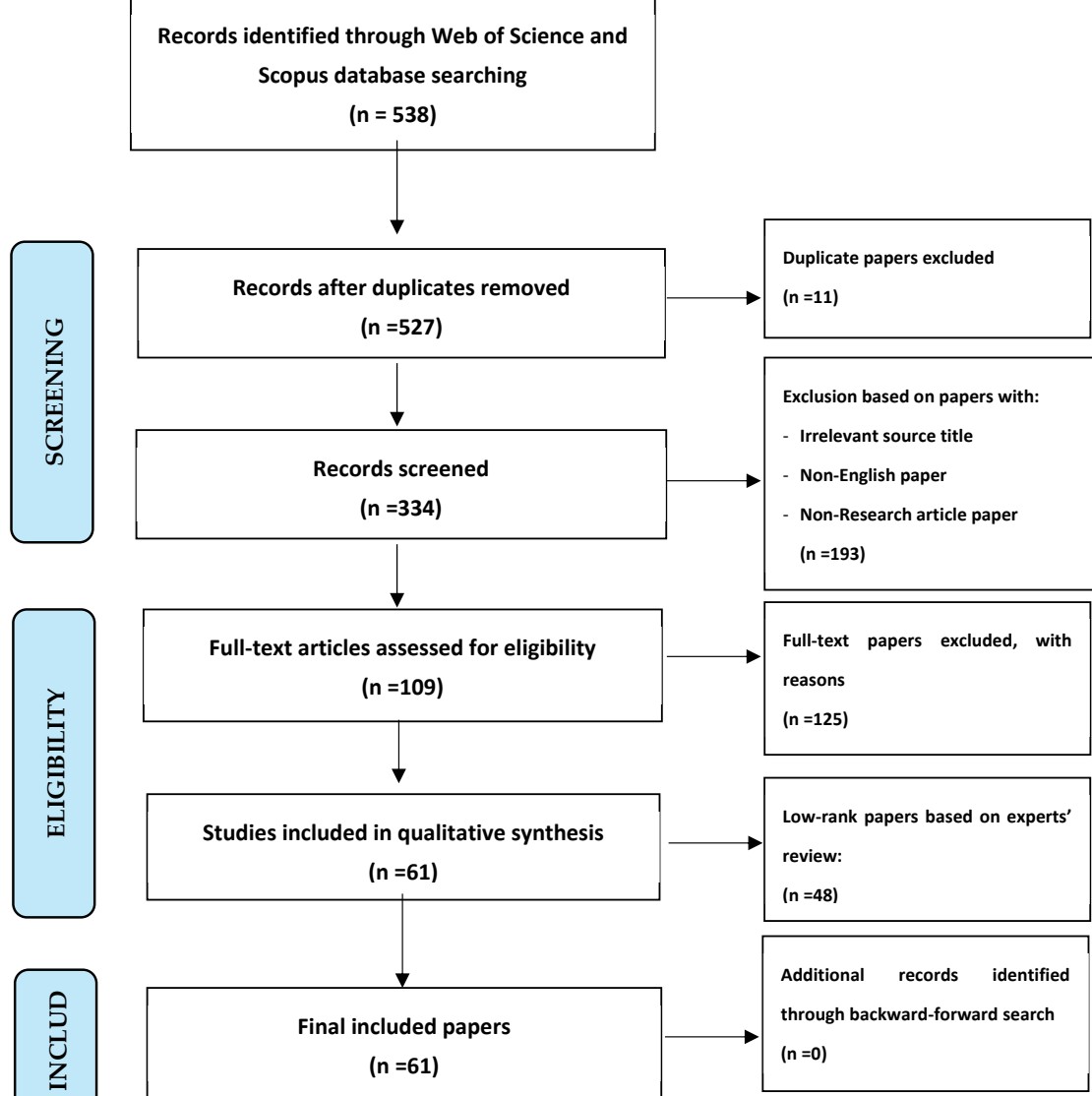

**Figure 1.** A flow diagram of the process.

### 2.3.1. Identification

Identification is the process of searching for the right keywords based on the research questions. In this study, three keywords and their synonyms, related terms, and variations were used. The keywords were <local wisdom>, <indigenous people>, and <nature conservation>. The keywords were developed based on the research question as suggested by Okoli [21], and the identification process relied on online thesauruses, keywords used by past studies, keywords suggested by Web of Science (WoS), and keywords suggested by experts. As shown in Table 1, this study managed to enrich a series of keyword strings using the WoS database and Scopus to search for the required articles. WoS and Scopus are websites that provide subscription-based access to multiple databases and have comprehensive citation data for many academic disciplines. The current researchers chose WoS and Scopus

rather than other search engines because all journals in the former underwent a thorough editorial process, thereby ensuring article quality before publication and maintaining the impact factor rankings of the journals [22,23]. The searching process using both databases resulted in 538 articles that were compatible with this study's research objectives.

**Table 1.** The search strings.

| Database | Search Strings |
|---|---|
| Web of Science | TOPIC: (Local wisdom OR local knowledge) Refined by: TOPIC: (indigenous people OR first people OR aboriginal people OR native people OR autochthonous people) AND TOPIC: (environmental conservation OR ecological conservation OR nature conservation) |
| Scopus | TITLE-ABS-KEY (local AND knowledge OR local AND wisdom) AND (indigenous AND people OR first AND people OR aboriginal) AND people OR native AND people OR autochthonous AND people) AND (environmental AND conservation OR ecological) AND (conservation OR nature AND conservation) |

### 2.3.2. Screening

All of the 538 articles were first screened to remove any duplicate records. A total of 11 papers were excluded from the record due to duplication, which left 527 articles remaining. The remaining 527 articles were screened again to ensure the quality of the review—only articles with empirical data and published in a journal were included. Moreover, only articles published in English were incorporated in the review to avoid confusion in understanding. This process excluded 193 articles as they did not fit the inclusion criteria. The remaining 334 articles were used for the third process—eligibility.

### 2.3.3. Eligibility

Eligibility is the third process, where the authors manually monitored the retrieved articles to ensure all the remaining articles (after the screening process) were in line with the criteria. This process was carried out by reading the title and abstract of the articles. A total of 125 articles were excluded due to a focus on local wisdom modeling, the local wisdom of the local community (not focusing on indigenous people), the adaptation of local wisdom in policymaking, the history of local wisdom, the philosophy of indigenous people, or a focus on review, not empirical data, or the methodology section was not clearly defined, or the articles were published in the form of a chapter in a book, a book, proceedings, or conference papers. Therefore, only 109 articles remained after this stage.

### 2.4. Quality Appraisal

In ensuring the quality of the content of the articles, the remaining articles were presented to two experts for quality assessment. As suggested by Petticrew and Roberts [24], the experts ranked the remaining articles into three quality categories, namely high, moderate, and low. Only articles categorized as high and moderate were reviewed. The experts focused on the methodology of the articles to determine the rank of the quality. Both authors had to mutually agree that the quality of the paper must at least be at a moderate level for it to be included in the study. Any disagreement was discussed between them before deciding on the inclusion or exclusion of the articles for the review. After this process, 44 articles were ranked as high, 17 as moderate, and 48 articles as low. Thus, only 61 articles were eligible for the review.

### 2.5. Data Abstraction and Analysis

In this study, thematic analysis was used in order to generate the themes and sub-themes. According to Braun and Clarke [25], thematic analysis is used to identify the themes and sub-themes based on efforts related to noting patterns and themes, clustering,

counting, and noting the similarities and relationships that exist within the abstracted data. Any similar or related abstracted data were pooled in a group. In this study, after a thorough analysis, a total of 7 themes were developed, which were: (a) forest management, (b) flora and fauna conservation, (c) food security, (d) water management, (e) land management, (f) weather forecasting, and (g) others.

## 3. Results

### 3.1. Spatial and Temporal Analysis of Selected Articles

This study analyzed a total of 61 articles after a meticulous and neat method of selection. Based on Figure 2a, the country with the highest number of articles was Indonesia with 12 articles, followed by Canada (8), Australia (7), and China with 5 articles. The rest of the countries recorded less than five articles, as shown in Figure 2a. All of the articles were distributed across 25 countries; most of the countries were from the Asian, American, and African continents. Only one article from Europe (Spain) was recorded in this study.

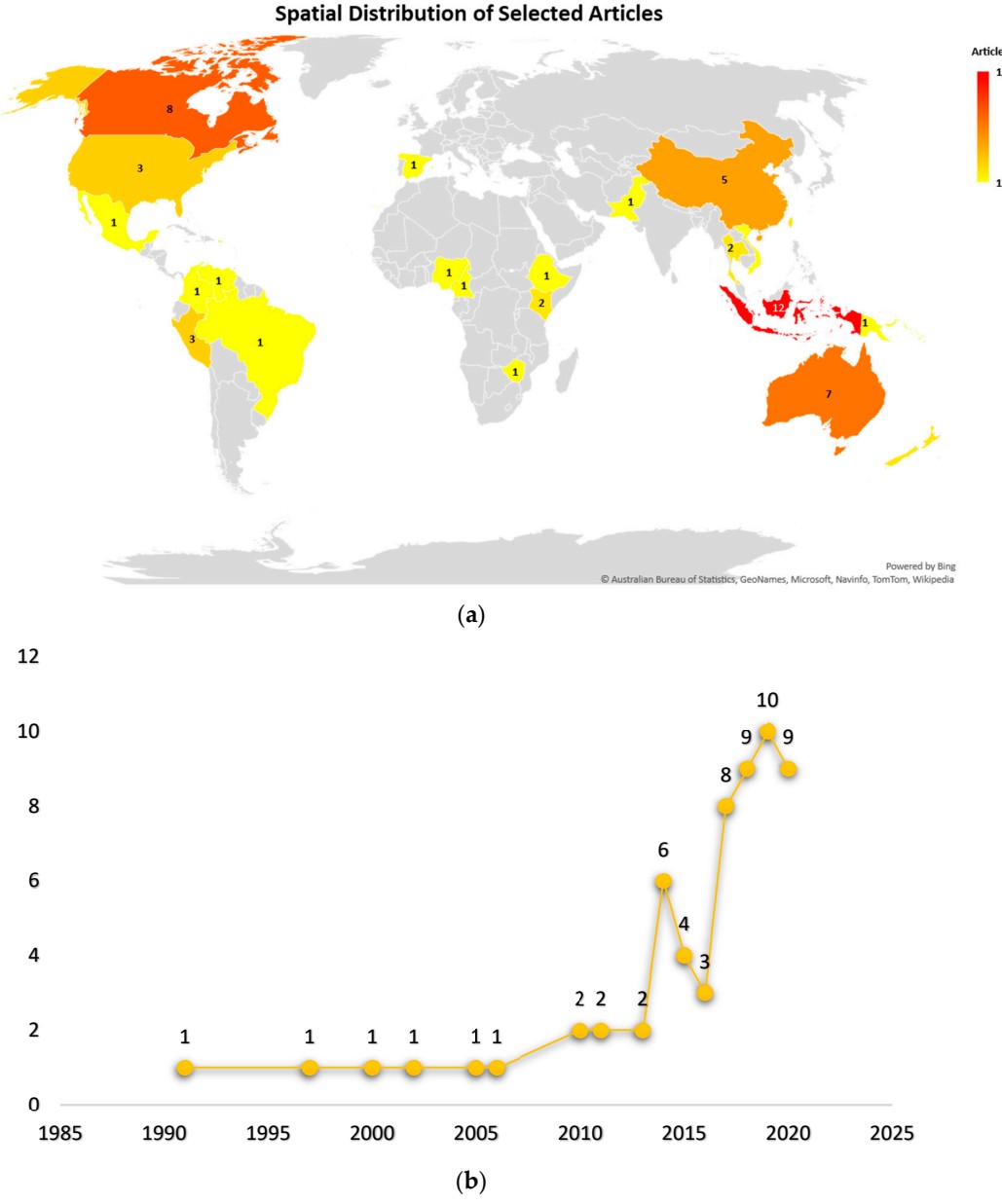

(**a**)

(**b**)

**Figure 2.** (**a**) The Spatial Distribution of Selected Articles. (**b**) The Temporal Distribution of Selected Articles.

Based on the performed analysis, the 61 articles were distributed starting from the year 1991 until 2020. The highest number was published in 2019 with 10 articles, followed by 2018 and 2020 with 9 articles, 2017 with 8 articles, 2014 with 6 articles, 2015 with 4 articles, and 2016 with 3 articles; the rest of the years had less than three articles. In the 1990s, only 1991 and 1997 have recorded articles related to this study. The trendline shows a significant increment in this area of study, especially between 2010 and 2020, and this may show the rise of interest in the subject from researchers and the significance of the issue to be tackled.

### 3.2. The Local Wisdom of Indigenous People in Nature Conservation—Contextual Issues

Three contextual issues were addressed from the analysis of the selected articles: (a) the type of method used (Figure 3a) and (b) the type of habitat (Figure 3b). Based on Figure 3a, there were four main methods used from the selected articles, with 59% (36 articles) using interviews as the main method to conduct the study on the local wisdom of indigenous people, followed by observation with 25% (15 articles), document analysis with 11% (7 articles), and survey with 5% (3 articles). Based on Figure 3b, there were nine types of habitats studied from the selected articles, with the most common being forests with 43% (26 articles), followed by grassland with 20% (12 articles), islands with 13% (8 articles), rivers and coasts, both with 6% (4 articles), respectively, mountains with 5% (3 articles), deserts with 3% (2 articles), and lastly tundra and mangrove with 2% (1 article) for both habitats.

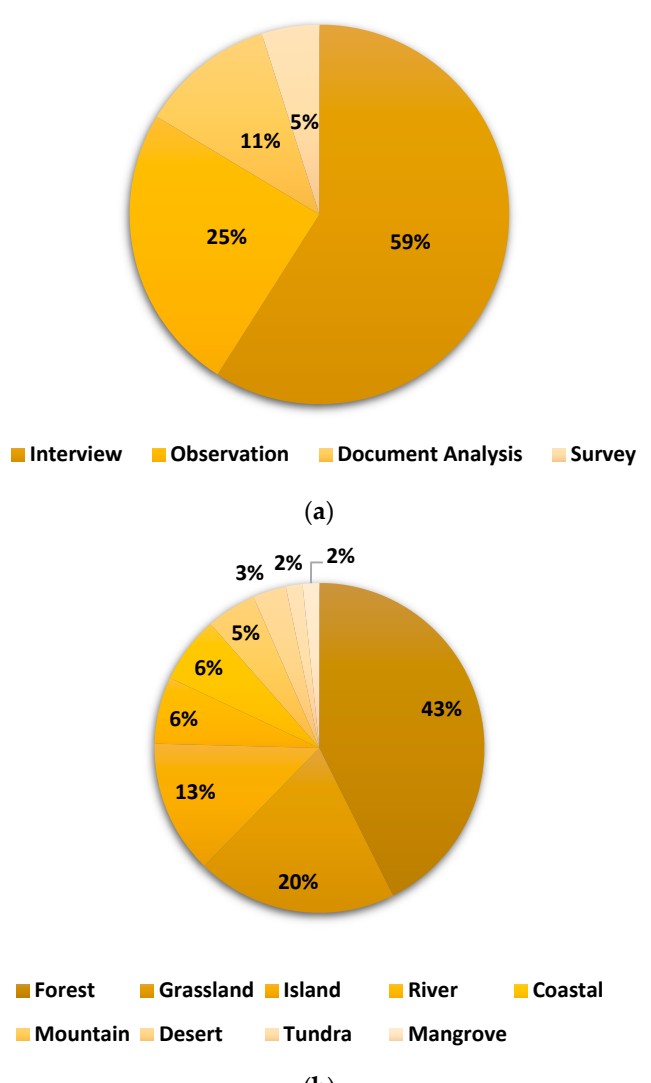

**Figure 3.** (**a**) Method used in Selected Articles. (**b**) Type of Habitat.

### 3.3. The Local Wisdom of Indigenous People in Nature Conservation—Thematic Analysis

A total of seven themes of the local wisdom of indigenous people in nature conservation were extracted from all of the 61 articles: (a) forest management, (b) flora and fauna conservation, (c) food security, (d) water management, (e) land management, (f) weather forecasting, and (g) others. Based on Figure 4, the most studied local wisdom of indigenous people was forest management with 21%, followed by flora and fauna conservation with 18%, food security with 15%, water management with 13%, land management with 12%, and weather forecasting with 8%; meanwhile, the other themes totaled 16%.

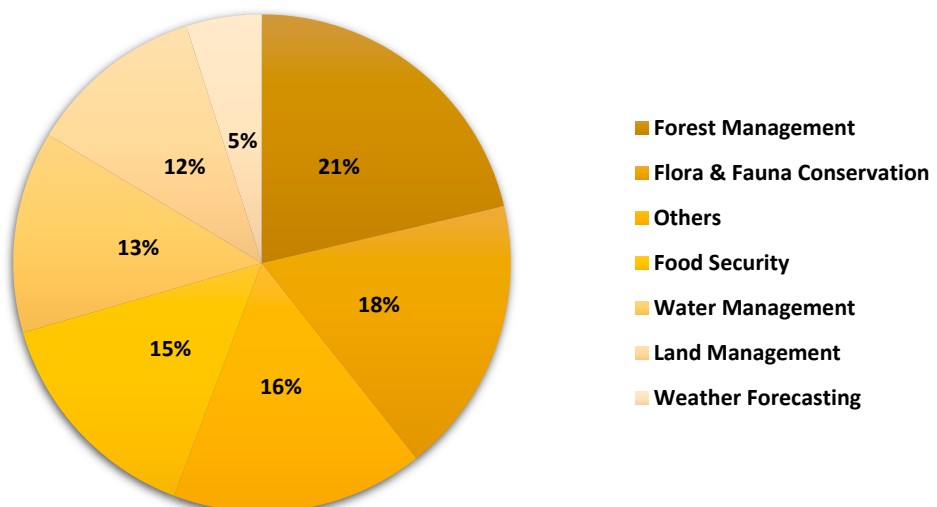

**Figure 4.** The Local Wisdom of Indigenous People in Nature Conservation.

(a)   Forest Management

Forests are the most important resources for indigenous people, especially those living in tropical ecosystems. Therefore, managing the forest resources is very important for the survival of the indigenous people who dwell in or depend on the forest. For example, the indigenous community in Indonesia believe that the forest is a sacred place that was granted by a divine element and is protected by their ancestors' spirits. According to Al Muhdhar et al. [26], the Tobaru Community applied three methods that were inherited through generations, Sasi, Kasse tanda, and Cincang. The conservation values of the Sasi method are religious-based plant protection, utilization, and management. The community believes that if the Sasi method is in place on their farm, God will protect and increase their farm yields and reduce the attacks of various diseases. Kasse tanda, in the tradition of the Tobaru traditional community, aims to improve the number of fruits grown and to enlarge their size. In addition, the Kasse tanda method could repel pests that, for example, will damage the vegetative organs of plants, such as the leaves and stems, and the generative organs, such as the flowers, fruits, and seeds. The Cincang method is a stem incision method. The function of the method is to reduce coconut bud rot and nutfall diseases. The method is very effective and still practiced now. Another tribe of indigenous people in Indonesia, the Kajang tribe, believe that the forest is a customary law that must be respected—according to them only the eldest with sufficient knowledge can determine which trees can be cut [27]. Other than that, indigenous people in Kenya will only cut down one tree for every three trees in an area [28]. Fache and Moizo [29] reported that the indigenous people in Australia plant trees at a certain distance to prevent bushfires from expanding quickly during the drought season. According to Rodriguez [30], the Pemon Tribe do not use fire in their forest due to the risk of the forest catching fire. Schulz et al. [31] discovered that the Urarina distinguish between ecosystems according to vegetation physiognomy, certain (palm) tree species, hydrology, and soil appearance and that their use of natural resources varies between different ecosystems. The indigenous people in China have practiced that

even without specific punishment measures—local people effectively controlled the use of natural resources through moral constraints, public-opinion constraints, and worship rituals. Furthermore, they formed a forest- and wildlife-protection system with banyan trees and the Derbyan parakeet at the core [32,33]. On the other hand, according to Ungirwalu et al. [34], the adaptive resource management of Black Fruit by the Wandamen people of Indonesia is based on an approach known as "ethnotechno-conservation". This approach is an attitude of mind by which Wandamen communities manage their Black Fruit trees to meet the dual goals of fulfilling subsistence needs and conserving the resources. According to Utomo et al. [35], the local knowledge in angklung (a musical instrument made of bamboo in Indonesia) making includes selecting benel (*Gigantochloa atter*) and ori (*Bambusa arundinacea*) as angklung materials, the age of the bamboo (more than three years old), the best month for logging between August and October, and air drying the bamboo without exposing it to direct sunlight and in an upright position.

(b)　Flora and Fauna Conservation

　　Besides forests, the flora and fauna that exist in the indigenous people's surroundings are also considered as divine and sacred. For example, the indigenous people in the USA have a unique way of conserving the salal (a shrub) [36], and other indigenous people, also in the USA, taught their children harvesting skills focused on small birds, and the adults also occasionally harvested shorebirds, but the shorebirds were not the primary food or cultural resource [37]. They will pick fern, huck, and brush for floral greenery for landscaping as substitutes for salal. They only harvest an area after 2–3 years and only harvest it at 15 years of age and above for good commercial salal. Meanwhile, the indigenous people in Canada only kill adult bears during the hunting season, which is around June–July [38]. According to Gilmore et al. [39], the Maijuna Tribe in Peru use the technique of tracing an animal's mineral licking behavior to monitor the animal's presence and the numbers of individuals. On the other hand, according to Franco and Minggu [40], the Iban tribe in Brunei provided additional locale-specific information on the dietary preferences, abundance, and conservation threats of hornbills to scientific knowledge and conservation. Meanwhile, the indigenous people in New Zealand know that swans are important for distributing fertilizer for their crops; therefore, they only hunt swans that are away from their nest to avoid overhunting [41]. Plus, the Tuawhenua from New Zealand understand the importance of the Kereru (pigeon) in the human–environment relationship [42]. According to Telfer and Garde [43], in western Arnhem Land, central northern Australia, the indigenous peoples know much about the rock kangaroos of the region, an animal lacking in any scientific data. Information about the ecology of the species is required for their conservation and management. Ethnoecological studies can assist senior indigenous people with the transfer of knowledge and can give respect and meaningful employment to those involved. Most of the indigenous people appreciated the flora and fauna due to its sacred value [44,45].

(c)　Food Security

　　For all living things, food is the most important source of life, an essential for survival. The same goes for indigenous people; in order to survive, they need to eat, they need to have a sustained supply of food. Therefore, indigenous people have developed specific wisdom based on their local habitat to ensure they have sufficient food. According to Adams et al. [46] and Gill and Lantz [47], the indigenous people in Canada only fish to feed their family; only medium and large fish were caught. Plus, no bombing or net methods were used, only traditional ways of fishing using spears of sharpened wood. The indigenous people in Mexico have been practicing crop rotation techniques for several decades to have various types of food available and decrease their dependency on the local staple food [48]. On the other hand, for the Tayal (indigenous people of Taiwan), migration is a common and communal activity, and more importantly, it is very possibly a result of a lack of food supplies under the impact of climate change. The Tayal practice fire-fallow cultivation because they do not separate forestry from agriculture as modern

land management would [49]. According to Kaewploy et al. [50], the indigenous people of Thailand establish their own or communal ponds. The ponds are surrounded by a combination of nipa palm (*Nypa fruticans*) and mangrove forest trees (*Rhizophora mangel*, red mangrove) and some coconut trees (*Cocos nucifera*) growing intensively for over 20 years. As observed, the local people practice some traditional rituals of respecting and protecting the mangrove areas, especially during the culture season, enabling them to seek abundance and blessings from the "so-called spirits/unseen" of the natural order. According to Torres-Vitolas et al. [51] and Parraguez-Vergara et al. [52], the indigenous people of Colombia and Peru have already converted their way of producing food from collecting natural resources to subsistence farming. This situation has also happened among the indigenous people of Australia where they have planted wheat and barley to secure their food supply [53]. The indigenous people of Nigeria have already used of a wide variety of plants in their diet to avoid malnutrition [54].

(d)   Water Management

Water is essential for any living thing on earth, especially for humans, as 60% of the human body is composed of water. Therefore, it is very important for indigenous people to conserve their water resources to ensure they always have sufficient clean water for drinking and other activities. According to Chunhabunyatip et al. [55], for the indigenous people of Thailand the upstream river cannot be touched because it is protected by the goddess who brings good fortune and prosperity. They only fish and use water from the downstream river. Meanwhile, the indigenous people of Indonesia practice the Aia Adat (water resources controlled and regulated by custom), which is one of their strategies to distribute the water. The general rule is that irrigation will flow from 6 p.m. to 6 a.m., regularly to all farmland, under the supervision of kapalo banda. When rains occur, water resources can be used during the day without special supervision [56,57]. The indigenous people of China created a very comprehensive water resource management and utilization system for the ponds of the chongchong paddy fields to confront the seasonal flooding and water shortages in their area [58]. On the other hand, the indigenous people of Australia have been practicing rainwater harvesting and digging wells to fulfil their water supply needs [59]. In addition, in northern Australia, the indigenous people from the remote community of Ngukurr have raised concerns about drinking water from freshwater billabongs due to potential microbial contamination from feral ungulates (buffalos, pigs, horses, and cattle). Boiling water when drinking from billabongs during all seasons is considered best practice to avoid ingestion of infective enteric pathogens [60]. At the other hand, according to Nguyen [61], for the Tay and Thai people in northern Vietnam water resources used for daily living activities are divided into two categories: water for drinking and water for other activities (bathing, washing, running rice mortars, etc.). Drinking water originates from upstream and/or from springs and is believed to be more hygienic than water from downstream, which is for bathing and washing. Thai people have several specific local customary laws that relate to the protection and management of water resources. They have lists of prohibitions against activities that are harmful to water for drinking, including grazing, burying the dead, releasing toxic contaminants (thuốc cá—a poison made from a species of forest tree, lime, and ash), defecating, discarding dead carcasses, slaughtering cattle and poultry, and so forth. They also have hit khoong to protect sacred forests which are watershed areas. Any violation will result in a fine ranging from several (an old unit of money) to three silver bars, together with meat and wine.

(e)   Land Management

Most of the known indigenous people dwell on land. They build their houses, huts, or shelters on land. It is important for them to take care of their land, not only because they live on it, but also because it provides food and other natural resources for them. According to Davies et al. [62], the indigenous people of Australia did not cut or burn trees in unnecessary areas whenever they explored new resettlements. In addition, the indigenous people of Australia also use of fences to create frames for feral ungulate management [63]. The

indigenous people in Spain believed that a specific area size can only accommodate a specific number of individuals [64]. On the other hand, the indigenous people of Indonesia (Outer Baduy) have learned to classify the soils based on color, water content, stoniness or rock parent material, and humus content. To maintain soil fertility in the swidden cultivation, the Outer Baduy people have developed some strategies, such as determining appropriate fallow time periods, applying zero tillage, and planting legume crops in both the swidden fields and the fallow land. Traditionally, because the Outer Baduy are forbidden to use inorganic fertilizers, the length of the fallow period and the kind of vegetation succession have an important role in maintaining soil fertility [65].

(f)    Climate Forecasting

The indigenous people are not usually equipped with modern technology, especially technology that involves remote sensing and satellites. In order to survive in a rapidly changing climate, the indigenous people need to use any information that they can receive from their surroundings for agriculture, hunting, building, and daily routines. The indigenous people in the Solomon Islands have been predicting disasters such as tsunamis or earthquakes by observing the behavior patterns of benthos [66], and this is also in agreement with the study by Garcia-Quijano [67], which has found that the indigenous people of Puerto Rico have learnt how to predict tsunami by looking at specific marine species assemblages. Meanwhile, according to Chen and Cheng [68] and Lin et al. [49], the indigenous people in Taiwan have been observing the flying patterns of birds and the existence of fish near the seashore as early alarms for tsunamis and heavy storms. The indigenous people of Indonesia have learnt on how to predict the climate that gives a significant impact towards their farming activities by watching the cloud formation [69,70].

(g)    Others

Other than the six categories that are mentioned above, there are several types of local wisdom used by the indigenous people that have been recorded from this study, such as ecosystem services, medicinal plant use, highland conservation, and built design. According to Carson et al. [71], the Baka tribe in Cameroon are the best traditional doctors. They know which plants can be good medicine for common diseases. In India, the indigenous people that possess knowledge of herbal healing are more committed than other villagers to preventing or mitigating the overharvesting of natural resources [72]. In terms of ecosystem services, the indigenous people in Ethiopia mostly appreciated a few services of high market value while most ecosystem services are not traded in local markets and hence not highly valued. Some low-rated ecosystem services, such as fodder and medicinal plants, were nonetheless widely used, demonstrating the need to also conserve low-rated ecosystem services that are used universally [73]. However, according to Shah et al. [74], the indigenous people in Pakistan are involved in highland conservation because the highlands symbolize power and longevity in their beliefs. Built design is very important for the indigenous people in Indonesia in mitigating disasters, such as landslides, drought, or pollution. This society has realized that disaster will occur if their environment is damaged. Local wisdom in disaster mitigation can be seen in the form of architecture, land use zoning, and land management for a sustainable environment [75,76].

## 4. Discussion

The local wisdom of the indigenous people was built and formed on the cultural values and the environmental dependency of the indigenous people. This also significantly affects the way indigenous people take care of their surroundings, which eventually leads to the good practice of nature conservation. In modern nature conservation, it is important for people to gain knowledge and useful information as the output from education. However, most indigenous people did not have access to modern education, or they chose not to use it due to their cultural beliefs [77]. Therefore, it is hard to analyze the local wisdom from indigenous people through a quantitative approach. Out of 61 papers, only 3 papers used surveys as the method to analyze the local wisdom of the indigenous people. Meanwhile,

the rest of the papers used interviews, participant observations, and document analysis. Indigenous people usually dwell in very remote areas, such as rainforests, vast grasslands, islands, montane forests, and deserts [78,79]. It is very seldom that indigenous people are found living in a modern society such as urban and rural areas. Therefore, that is why most of the papers analyzed have studied remote areas such as forests, grasslands, mountains, and islands. The first paper in this systematic review study was published in 1991, and the number of published papers has continued increasing from 2010 until now. This shows that even though this study was conducted starting from decades ago, the interest coming from researchers from all over the world is still increasing. The increment of interest gives a sign that the study of the local wisdom of indigenous people in conserving nature is still relevant; plus, there are a lot of things that remain unanswered [80]. In terms of spatial distribution, most of the papers were from countries known to have large populations of indigenous people, such as Indonesia, Australia, and Latin America. However, it is really surprising that there was a smaller number of studies conducted from Southeast Asian countries, African countries, and other Asian countries. This may lead to insufficient knowledge and may have an effect on the overall understanding towards the local wisdom of the indigenous people in nature conservation in the world [81].

The forest is the most important element in indigenous people's beliefs. Everything about the forest, according to Carson et al. [71], is sacred and divine to the first people. It is a belief passed down through generations of indigenous people. They believe that the forest is the essence of life, that it provides them with everything they need, including food, water, a place to live, and spiritual strength [82]. For the indigenous people, the tree is the most revered and sacred of all living things in the forest. The indigenous people referred to trees as "mothers", and it is critical to care for the trees in the forest. Trees will be cut down in accordance with indigenous peoples' customary law, which states that only mature and large trees will be cut down. According to Schmidt et al. [83], it is also important in modern logging to determine the size and age of a tree before it is cut down. This is done to ensure the long-term viability of timber production as well as the conservation of the forest ecosystem [27]. Furthermore, indigenous people are skilled at forest replantation strategies such as selecting unique species that will aid them and the forest in surviving natural disasters such as drought, flood, and storms. According to Armstrong et al. [84], indigenous peoples recognized the regulation service provided by certain species long before modern society.

The indigenous people rely on flora and fauna for food, minerals, and other necessities. Aside from that, some animals and plants were revered by indigenous peoples. According to Aswita et al. [44], the indigenous peoples of North America regard bison as a sacred and powerful giver of life. During ceremonies, their horns and hides were used as sacred regalia. Plants such as sage, tobacco, sweetgrass, and red cedar are also considered sacred due to their versatility of use [45]. This is consistent with the findings of Franco and Minggu [40], who discovered that the indigenous people of Brunei considered the hornbill to be sacred, believing that its presence in the forest would bring good fortune and peace to the community and the forest. As a result, they have learned to track and identify the hornbill's presence based on its eating habits and leftovers. The indigenous people have also inherited a systematic hunting method from their forefathers. According to Agatha [85], indigenous people only fish or hunt adult animals for survival. This will allow the animal to reproduce indefinitely while also maintaining the natural ecosystem's balance [42].

Food security is the most concerning issue, not only for indigenous peoples, but for all of humanity. Food security and sustainability are critical for indigenous peoples' survival. The indigenous people usually obtain their food from their immediate surroundings. They hunt, collect, and grow food as a community to avoid exhaustion and to maximize food production [86]. This is also referred to as the indigenous peoples' food system. According to Ibarra et al. [48], indigenous peoples have used crop rotation to ensure they have enough food all year. This will also assist the soil in remaining fertile. There are also

indigenous people who are already practicing mix farming to increase food production in their community. This method will not only save time, but it will also require fewer human resources [49].

Water, according to indigenous people, is a sacred gift that connects all life. All indigenous people place a high value on water and perform sacred ceremonies to ensure that water is respected and that these water ceremonies are passed down to future generations [56]. There are several indigenous practices that are considered to be methods for managing water resources, such as irrigation scheduling, strategic water consumption, and upstream river conservation [55]. The irrigation scheduling, according to Chief et al. [87], could benefit both indigenous people and the ecosystem. This method can ensure the sustainability of their water resources while also preventing overflowing, which could lead to flooding. They use of strategic water consumption methods, such as boiling water for drinking, demonstrates their understanding of the dangers of untreated drinking water [59]. Aside from that, rainwater harvesting is another way for them to reduce the burden placed on the river in terms of water supply [57]. Essentially, the upstream river was regarded as divine and held great spiritual significance for the indigenous people, particularly the Asian indigenous people. As a result, they rarely fish or bathe in the upstream to avoid the wrath of the water spirit. However, in modern knowledge, this practice is defined differently, where avoiding using upstream river water reduces the discharge of pollution from the upstream river, which is then accumulated in the downstream river.

Land is important to many indigenous people in all aspects of their lives, including culture, spirituality, language, law, family, and identity. Rather than owning land, each person is related to a piece of land through the kinship system. According to Schultz et al. [88], indigenous people have learned to identify soil profiles using the wisdom passed down from their forefathers. This wisdom can also be used by modern society to observe their surroundings and act as an early warning system to indicate the soil quality of their land [63]. The indigenous people also consider any large tree on a piece of land to be sacred. As a result, whenever they need to clear land for new settlers, they avoid felling the large tree. However, according to modern knowledge, this act can help maintain the strength of the land due to the bind by large tree roots, as well as avoid land slide [65]. The indigenous people's nomadic way of life has resulted in the development of weather forecasting wisdom. They must forecast the weather in order to plan and strategize their travel and farming routines [89]. Some indigenous people are wise in observing cloud formation as well as wind direction and strength. This will help them predict the rainy season and avoid unneeded casualties. Aside from the atmosphere, indigenous people can predict tsunamis and typhoons from the sea by observing fish swimming patterns and benthos formations. The swimming pattern of fish at the seashore, according to Mojo et al. [69], can be an indicator of a tsunami from the sea. As a result, indigenous peoples can take early action to mitigate the greater impact of the tsunami and typhoon on their community.

## 5. Conclusions

The present study reviewed 61 papers on the local wisdom of indigenous people in nature conservation to gain insight into how indigenous people all over the world used their local wisdom to directly or indirectly conserve nature. This study also revealed the current scenarios and trends on how researchers have studied the local wisdom of indigenous people, where the number of papers published in this area of study has been increasing from 1991 until now. The papers are also well distributed among countries with known indigenous populations. However, there are also known countries with indigenous people that were not represented by any papers in this study, such as Malaysia, India, etc. Most studies focus on remote types of habitat, such as rainforest, grassland, mountain, and desert. Due to many indigenous peoples' use of oral communication and history, most of the studies were conducted using qualitative approaches, such as interviews, observations, and document analysis. Through content analysis, this study can be divided into seven main categories: (a) forest management, (b) flora and fauna conservation, (c) food security,

(d) water management, (e) land management, (f) weather forecasting, and (g) others. All these discoveries show that the indigenous people have a very practical and efficient way of conserving the nature which can be adapted to the existing methods. It is time to learn from those who have assimilated with nature for generations to ensure that the future of our nature is assured.

This systematic review paper confirms that there were several limitations and gaps in the study of the local wisdom of indigenous people in nature conservation. Firstly, information regarding the local wisdom in conserving nature from indigenous people is lacking from some of the countries known to be populated by indigenous people. Moreover, most of the studies only focused on how indigenous people practice their local wisdom daily for nature conservation, but seldom studied how this wisdom could be adapted towards a scientific approach. Plus, for future studies, it is recommended that researchers explore a few countries such as Malaysia and India to analyze the local wisdom of the indigenous people of these countries. Lastly, an emphasis on a framework to adopt the local wisdom of indigenous people in nature conservation into the scientific approach, such as the use of technology in the improvisation of experimental designs, would be beneficial.

**Author Contributions:** Conceptualization, A.A. (Azlan Abas); methodology, A.A. (Azahan Awang) and A.A. (Azmi Aziz); software, A.A. (Azlan Abas); validation, A.A. (Azahan Awang); formal analysis, A.A. (Azlan Abas); investigation, A.A. (Azlan Abas); resources, A.A. (Azlan Abas); data curation, A.A. (Azlan Abas); writing—original draft preparation, A.A. (Azlan Abas); writing—review and editing, A.A. (Azmi Aziz); visualization, A.A. (Azmi Aziz); supervision, A.A. (Azahan Awang); project administration, A.A. (Azlan Abas); funding acquisition, A.A. (Azlan Abas). All authors have read and agreed to the published version of the manuscript.

**Funding:** This research has been funded by Universiti Kebangsaan Malaysia through research grant (SK-2020-024).

**Institutional Review Board Statement:** Not applicable.

**Informed Consent Statement:** Not applicable.

**Data Availability Statement:** Not applicable.

**Acknowledgments:** This study has been supported by Universiti Kebangsaan Malaysia through research grant (SK-2020-024). The authors also want to thank ReadByRose (Rose Norman) for providing an excellent proofreading service for this paper.

**Conflicts of Interest:** The authors declare that this study has no conflict of interest.

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
