# Peer review of "A Systematic Review on the Local Wisdom of Indigenous People in Nature Conservation"

_sustainability, doi:10.3390/su14063415_

Round 1
Reviewer 1 Report
Abas et al. Provide a timely assessment of the utility of IP knowledge and wisdom to improved nature conservation. This paper is an important contribution to the statement that Ips are the best guardians of the forest (and other ecosystems).
The authors also analysed a huge dataset of existing publications. However, there are lacks with the methodological approach, such as limiting to English language (which often includes many ecoregions). Given these restrictions the main criticism is that the conclusions are then based on a very limited data set of 61 publications.
I strongly recommend the authors to revise that decision has dramatically reduced the potential impact and conclusion and utility of the analysis. Given the existence of online translations software it is not too challenging to translate Spanish or French language publications. This would substantially fill a regional gap (e.g. francophone Africa).
Furthermore, the way the results are presented can be substantially improved. Tables and figure are often duplicates and not very insightful (and large descriptive). A summary table might be sufficient and can help to reduce a lot of text in the results section.
Then the authors have a better structure, and this will help the in discussing their results – where is importance, where are gaps, what is the impact of the study, for example for new conservations trends, such as 30x30 or OECMs.
Author Response
ANSWER TO REVIEWER’S COMMENTS
Reviewer 1
Abas et al. Provide a timely assessment of the utility of IP knowledge and wisdom to improved nature conservation. This paper is an important contribution to the statement that Ips are the best guardians of the forest (and other ecosystems).
Answer: We would like to thanks the reviewer for this comment.
The authors also analysed a huge dataset of existing publications. However, there are lacks with the methodological approach, such as limiting to English language (which often includes many ecoregions). Given these restrictions the main criticism is that the conclusions are then based on a very limited data set of 61 publications. I strongly recommend the authors to revise that decision has dramatically reduced the potential impact and conclusion and utility of the analysis. Given the existence of online translations software it is not too challenging to translate Spanish or French language publications. This would substantially fill a regional gap (e.g. francophone Africa).
Answer: We would like thanks the reviewer for his/her comment. However, the reason for choosing only English paper because of it is usually published on Web of Science and Scopus. From our previous searching, only 5 papers were found not written in English. Which mean, the exclusion of those papers will not bring down the quality of our analysis.
Furthermore, the way the results are presented can be substantially improved. Tables and figure are often duplicates and not very insightful (and large descriptive). A summary table might be sufficient and can help to reduce a lot of text in the results section. Then the authors have a better structure, and this will help the in discussing their results – where is importance, where are gaps, what is the impact of the study, for example for new conservations trends, such as 30x30 or OECMs.
Answer: Thank you for this comment. The authors presented all the data based on the issues that this paper need to address. Therefore, we chose to stick with our current data presentation.

Reviewer 2 Report
General comments:
This study examines examples of the use and conservation of natural resources by local communities in various countries around the world. Seven categories of natural resources relevant to the analyses are highlighted. The overall goal of this work was to identify traditional knowledge about environmental protection and compare its implementation among peoples in different regions of the world. It would also be useful to have an approach to the use of this kind of knowledge in modern methods of environmental protection, which was not included in this work.
A strength of the paper was the use of recent publications, most of which were concentrated in the 2015-2020 time frame. Only the top-and medium-rated papers were used. On the other hand, it resulted in a strong bias, as numerous papers on this topic are published in low-rated journals. A considerable amount of interesting and important knowledge from the perspective of environmental conservation was not included in this work. However, this is a necessary sacrifice in order to meet the methodological objectives, and therefore it is not recommended to change this status. The weakness of the document is the complete lack of alignment with editorial requirements. This is the case in the way citations are made, the order of papers in the literature list, and the way references are written. Detailed recommendations are presented later in the evaluation.
This review article contains a significant body of research on different regions of the world and different environments. Most studies covered areas from Southeast Asia. This is in line with the observations of other authors. The lack of systematization of studies on traditional ecological knowledge in nature conservation was identified as a knowledge gap, as well as the need for a designated trend and pattern for the study of this phenomenon. The work addresses this need to some extent.
Silmar Reviews has been performed in the last years, (for example Joa 2018). However, these works were focused mainly on one or a few types of environment. It was found that there is a lack of such studies from Europe, while they are common in Hungary (Biró et al. 2019). I do not know if these publications were identified and rejected at the verification phase, however, they seem to be relevant. It has also been claimed that there is a lack of analysis from within India, while such studies have been conducted (Rath et al. 2020). Please consider these items or justify why they are included.
Only a few, necessary self-citations were used in this article. In the matter of novelty, 49 of 61 analyzed positions were created between 2015 and 2020. This number only refers to works used in the results paragraph. In the whole document, 90 works have been cited. Only 31 of them were made before 2016, the remaining 59 were written in the last 5 years, including 4 works from 2021. These statistics show the high novelty of this work.
The results, discussion, and conclusion chapters follow each other very closely, which on the one hand supports the coherence of the paper, but on the other hand, gives the impression of constant repetition of the same content. The discussion brings little interpretation to the results, and there is a lack of suggestions for viable application of the identified knowledge in environmental protection.
The conclusions are too elaborate and not clear enough. It is recommended to reduce the volume of this paragraph and clarify the achievements of this work.
The overall readability of the paper is good, and there are no concerns about language correctness. However, there is a need to increase the readability of the figures presented, as described in the specific requirements.
The overall evaluation of the paper is positive, however, a major revision of the structure of the paper is recommended.
Specific comments:
21 Too many keywords. Key word should be limited to three, like in methodology in line 110
25-32 The word „human” is overused in this paragraph, please use synonyms such as people, mankind,
27 Incorrect wey of citation. In the text, reference should be represented by numbers in square brackets [ ]. A number of references in brackets should reflect the order in which this reference has been used. Apply to the whole document.
34 Citation and self-citation
36 Slow pace -> slow rate
106 Cited figure should be placed directly under this line. All Figures, Schemes, and Tables should be inserted into the text close to their first citation.
167 Based on the performed analysis, not on the figure
172 „Based on figure” again.
182 Figures should have titles.
198 It is recommended to increase the readability of figure 3b by mowing numbers up the center.
424 Unnecessary dot.
151-153 Continuous use of "indigenous people", use their or another reference if it is clear who we talk about.
540-750 References should be placed in order of their use in the text, tot in alphabetic order. For example, the first reference used in the text is:
Sneed, A. (2019). What conservation efforts can learn from Indigenous communities. Scientific American. May, 29. https://www.scien-tificamerican.com/article/what-conservation-efforts-can-learn-from-indigenous-communities/
And the last one is:
Mojo, E., Hadi, S. P., & Purnaweni, H. (2017). Sedulur sikep’s environmental wisdom in conservation of North Kendeng mountains Sukolilo. Advanced Science Letters, 23(3), 2504–2506.
Please use instructions for authors: https://www.mdpi.com/journal/sustainability/instructions#top
The reference number on the list is the same number that has been used in the text. If reference is used again later on in the text, the number of this reference stays the same.
References in the list are formated in incorrect wey. The order should be: Author(s), the title of the work, averted name of the journal, year, journal number, and a number of pages. See instructions for authors for MPDI journals.
Author Response
ANSWER TO REVIEWER’S COMMENTS
Reviewer 2
This study examines examples of the use and conservation of natural resources by local communities in various countries around the world. Seven categories of natural resources relevant to the analyses are highlighted. The overall goal of this work was to identify traditional knowledge about environmental protection and compare its implementation among peoples in different regions of the world. It would also be useful to have an approach to the use of this kind of knowledge in modern methods of environmental protection, which was not included in this work.
A strength of the paper was the use of recent publications, most of which were concentrated in the 2015-2020 time frame. Only the top-and medium-rated papers were used. On the other hand, it resulted in a strong bias, as numerous papers on this topic are published in low-rated journals. A considerable amount of interesting and important knowledge from the perspective of environmental conservation was not included in this work. However, this is a necessary sacrifice in order to meet the methodological objectives, and therefore it is not recommended to change this status. The weakness of the document is the complete lack of alignment with editorial requirements. This is the case in the way citations are made, the order of papers in the literature list, and the way references are written. Detailed recommendations are presented later in the evaluation.
This review article contains a significant body of research on different regions of the world and different environments. Most studies covered areas from Southeast Asia. This is in line with the observations of other authors. The lack of systematization of studies on traditional ecological knowledge in nature conservation was identified as a knowledge gap, as well as the need for a designated trend and pattern for the study of this phenomenon. The work addresses this need to some extent.
Silmar Reviews has been performed in the last years, (for example Joa 2018). However, these works were focused mainly on one or a few types of environment. It was found that there is a lack of such studies from Europe, while they are common in Hungary (Biró et al. 2019). I do not know if these publications were identified and rejected at the verification phase, however, they seem to be relevant. It has also been claimed that there is a lack of analysis from within India, while such studies have been conducted (Rath et al. 2020). Please consider these items or justify why they are included.
Only a few, necessary self-citations were used in this article. In the matter of novelty, 49 of 61 analyzed positions were created between 2015 and 2020. This number only refers to works used in the results paragraph. In the whole document, 90 works have been cited. Only 31 of them were made before 2016, the remaining 59 were written in the last 5 years, including 4 works from 2021. These statistics show the high novelty of this work.
The results, discussion, and conclusion chapters follow each other very closely, which on the one hand supports the coherence of the paper, but on the other hand, gives the impression of constant repetition of the same content. The discussion brings little interpretation to the results, and there is a lack of suggestions for viable application of the identified knowledge in environmental protection.
The conclusions are too elaborate and not clear enough. It is recommended to reduce the volume of this paragraph and clarify the achievements of this work.
The overall readability of the paper is good, and there are no concerns about language correctness. However, there is a need to increase the readability of the figures presented, as described in the specific requirements.
The overall evaluation of the paper is positive, however, a major revision of the structure of the paper is recommended.
Answer:
Specific comments:
21 Too many keywords. Key word should be limited to three, like in methodology in line 110
Answer: Thanks, we already updated the keywords to three.
25-32 The word „human” is overused in this paragraph, please use synonyms such as people, mankind,
Answer: We already updated the few of the word “human” into “mankind”.
27 Incorrect wey of citation. In the text, reference should be represented by numbers in square brackets [ ]. A number of references in brackets should reflect the order in which this reference has been used. Apply to the whole document.
Answer: We already corrected the referencing format according to MDPI format.
34 Citation and self-citation
Answer: yes, but it is significant to cite this reference for this sentence.
36 Slow pace -> slow rate
Answer: thanks, we already changed it.
106 Cited figure should be placed directly under this line. All Figures, Schemes, and Tables should be inserted into the text close to their first citation.
Answer:
167 Based on the performed analysis, not on the figure
Answer: We have changed the sentence as suggested. Thanks.
172 „Based on figure” again.
Answer: We have changed the sentence as suggested. Thanks.
182 Figures should have titles.
Answer: Title has been added to every figure. Thanks
198 It is recommended to increase the readability of figure 3b by mowing numbers up the center.
Answer: Thanks for the recommendation, we have changed the position of the data.
424 Unnecessary dot.
Answer: the dot has been removed.
151-153 Continuous use of "indigenous people", use their or another reference if it is clear who we talk about.
Answer: Thanks for highlighting this, we have changed the word accordingly.
540-750 References should be placed in order of their use in the text, tot in alphabetic order. For example, the first reference used in the text is:
Sneed, A. (2019). What conservation efforts can learn from Indigenous communities. Scientific American. May, 29. https://www.scien-tificamerican.com/article/what-conservation-efforts-can-learn-from-indigenous-communities/
And the last one is:
Mojo, E., Hadi, S. P., & Purnaweni, H. (2017). Sedulur sikep’s environmental wisdom in conservation of North Kendeng mountains Sukolilo. Advanced Science Letters, 23(3), 2504–2506.
Please use instructions for authors: https://www.mdpi.com/journal/sustainability/instructions#top
The reference number on the list is the same number that has been used in the text. If reference is used again later on in the text, the number of this reference stays the same.
References in the list are formated in incorrect wey. The order should be: Author(s), the title of the work, averted name of the journal, year, journal number, and a number of pages. See instructions for authors for MPDI journals.
Answer: We have updated the reference list and citing format according to MDPI format.

Round 2
Reviewer 2 Report
The required amendments have been made throughout the paper. There are no objections to other areas of this work.